# Capsaicin and 1,25-Dihydroxyvitamin D_3_ Glycoside: Effects on the Reproductive Performance of Hyper-Prolific Sows

**DOI:** 10.3390/ani13172794

**Published:** 2023-09-02

**Authors:** Julie Gabriela Nagi Dario, Eduardo Raele de Oliveira, Rodrigo Pereira de Souza, Sabrina Theodorovicz, Giovana Chimentão Bernini, Gabriela Ruiz, Rafael Humberto de Carvalho, Caio Abércio da Silva

**Affiliations:** 1Animal Science Program, Center of Agrarian Sciences, State University of Londrina, Londrina 86057-970, PR, Brazil; gabriela93nagi@gmail.com (J.G.N.D.); sabrinatheodorovicz@hotmail.com (S.T.); giovanacbernini@hotmail.com (G.C.B.); gabriela.ruiz@cvale.com.br (G.R.); rafael.carvalho@uel.br (R.H.d.C.); 2Project Coordinator and Assistant at NutriQuest TechnoFeed, São Paulo 13025-320, SP, Brazil; eduardo.raele@hotmail.com (E.R.d.O.); rodrigosouza8@hotmail.com (R.P.d.S.)

**Keywords:** additives, dystocia, maternity, piglets, vitamin D

## Abstract

**Simple Summary:**

To investigate the potential benefits of phytogenic additives in improving the reproductive performance of sows, we evaluated the effects of capsaicin (CAP) and vitamin D3 [1,25-(OH)_2_D_3_], obtained from *Capsicum* spp. and *Solanum glaucophyllum*, respectively. Our experiments examined the effects of these additives when used in combination or individually. The results of our study indicate that supplementing diets with vitamin 1,25-(OH)_2_D_3_ can reduce the duration of farrowing, stillbirths, and dystocia rate. Colostrum production was also significantly higher in sows that received Vit 1,25-(OH)_2_D_3_ supplementation. Furthermore, the use of these additives contributed to litter growth and reduced the incidence of diarrhea in piglets. These findings suggest that CAP and Vit 1,25-(OH)_2_D_3_ could serve as valuable tools to improve the reproductive performance of sows.

**Abstract:**

This study evaluated the effect of a natural source of vitamin D_3_ [1,25-(OH)_2_D_3_] and capsaicin (CAP) in the dietary supplementation of sows in the final phase (85–114 days) of gestation (Gest) and lactation (Lact) on the reproductive performance of the sows and health of piglets through two experiments (Exp I and II). In Exp I, 120 sows were subjected to four treatments: T1—control (without [1,25-(OH)_2_D_3_] and supplemental CAP); T2—3.5 µg 1,25-(OH)_2_D_3_/Gest/day and 7.0 µg Vit 1,25-(OH)_2_D_3_/Lact/day; T3—7.0 µg CAP/Gest/day and 14.0 µg CAP/Lact/day; T4—1.75 µg Vit 1,25-(OH)_2_D_3_ + 3.5 µg CAP/Gest/day and 3.5 µg 1,25-(OH)_2_D_3_ + 7.0 µg CAP/Lact/day. In Exp II, 200 sows were randomly blocked, factorial 2 × 2 (without or with Vit 1,25-(OH)_2_D_3_ and without or with CAP): T1—control (without Vit 1,25-(OH)_2_D_3_ and CAP); T2—3.5 µg Vit 1,25-(OH)_2_D_3_/Gest/day and 7.0 µg Vit 1,25-(OH)_2_D_3_/Lact/day; T3—7.0 µg CAP/Gest/day and 14.0 µg CAP/Lact/day; T4—3.5 µg Vit 1,25-(OH)_2_D_3_ + 7 µg CAP/Gest/day; and 7.0 µg Vit 1,25-(OH)_2_D_3_ + 14.0 µg CAP/Lact/day. The duration of delivery (3:48 vs. 4:57 h) and the percentage of stillbirths (5.37% vs. 7.61%) were improved (*p* < 0.05) in the group that received Vit 1,25-(OH)_2_D_3_ (Exp II) compared to the control group. Moreover, the dystocia rate decreased (*p* < 0.05) in Exp II, which received Vit 1,25-(OH)_2_D_3_ (4.21 vs. 27.63%), and in Exp I, which received the combination of Vit 1,25-(OH)_2_D_3_ + CAP (12 vs. 40%) compared to the respective control groups. Colostrum production was greater (*p* < 0.05) in sows that received Vit 1,25-(OH)_2_D_3_ supplementation compared to the control group, consequently resulting in higher colostrum intake (*p* < 0.05) of the piglets (330 vs. 258 g/piglet). The additives reduced the incidence of diarrhea (*p* < 0.05) in piglets (Exp I and II). Thus, the use of additives improved the reproductive performance of sows and contributed to litter growth.

## 1. Introduction

The high reproductive rate of commercial sows results in lower birth weight and less uniform piglets at birth [1]. Additionally, these sows have a prolonged farrowing duration, necessitating increased physical effort and energy demands [2]. This leads to variations in plasma glucose levels, consequently contributing to an elevated stillborn rate and reduced piglet vitality [3].

Longer farrowing lengths in sows may increase the calcium demand of the sow, which might exhibit hypocalcemia [4]. In line with this finding, hyper-prolific sows present electrolyte homeostasis deficiencies (Ca, Mg, and P) around and during parturition, reflecting the large number of dystocic farrowings [5]. Large litters also have limited colostrum intake, impairing passive and active immuno-protection and predisposing them to diseases and poor performance [6].

Hyper-prolific sows have recognized high and specific nutritional requirements [7], and some recent experiments with the use of phytogenic additives aimed at meeting these demands have shown good results. In this group of additives, capsaicin obtained from pepper (*Capsicum* spp.) has known flavoring properties with effective gastroprotective and antioxidant activities [8]. It also stimulates the production and secretion of bile acids by the liver [9].

Capsaicin, when combined with other plant extracts, has been associated with the replacement of growth-promoting antibiotics in animal production, stimulating appetite, improving digestion, and acting as a modulator of the intestinal microbial ecosystem [10]. The use of these compounds increases immune status through the control of intestinal intraepithelial lymphocytes and via potent anti-inflammatory effects that reduce gastrointestinal challenges [8].

Conversely, vitamin D, whose main role is calcium and phosphorus homeostasis involved in several metabolic processes [11], has an extremely limited presence in plants; however, with some exceptions, vitamin D is found in the species *Solanum glaucophyllum*, *Solanum malacoxylon*, *Cestrum diurnum,* and *Trisetum flavescens* [12] in the form of glycosylated 1,25-dihydroxyvitamin D_3_ (vitamin D_3_) [13], a water-soluble molecule with high stability and bioactivity, compared with calcitriol [14].

In animals, the concentrations of vitamin 1,25-(OH)_2_D_3_ in tissues depend on its content in the diet and/or exposure to sunlight. During gestation and lactation, there is a significant increase in calcidiol in the blood plasma (the main circulating form of vitamin D). This condition indicates a high demand for calcium in these important moments involving reproduction [15], highlighting the role of vitamin D_3_.

Identifying natural additives that can meet the demands of the reproductive phases of sows is compatible with trends in the segment. This confers greater welfare and productive efficiency to sows and their litters. Several previous studies have highlighted the individual effects of capsaicin and vitamin D_3_ [1,25-(OH)_2_D_3_] supplementation in sow diets, suggesting that these phytogenic additives have the potential to improve reproductive performance. Based on these findings, we hypothesize that the use of these additives, either individually or in combination, can lead to positive outcomes in sow reproductive performance. Therefore, the objective of this study was to evaluate two plant-based dietary additives, capsaicin and vitamin 1,25-(OH)_2_D_3_, administered via feed, combined or not, to sows from day 85 of gestation to weaning, on reproductive performance and piglet health.

## 2. Materials and Methods

This experiment was conducted under the approval protocol of the Ethics Committee on Animal Use of the State University of Londrina (Universidade Estadual de Londrina—CEUA/UEL), number 060/2020.

The source of 1,25-dihydroxyvitamin D_3_ glycoside used in this study was the commercial product Panbonis^®^ (10 mg of 1,25-dihydroxyvitamin D_3_ glycoside/kg), which is the base of the *Solanum glaucophyllum* plant. Capsaicin, an active ingredient extracted from Malagueta pepper (*Capscicum frutescens),* originated from the commercial product Capsin^®^.

The evaluations were conducted in two commercial farms, with capacities of 2000 (Experiment I) and 6000 sows (Experiment II). Both farms did not have air conditioning systems in the gestation and lactation phases.

The farms were free of Porcine Reproductive and Respiratory Syndrome (PRRS) and Porcine Epidemic Disease (PEDv), and the sows received the vaccine Porcilis^®^ Ery+Parvo+Lepto (MSD Animal Health) 30 days before mating.

The evaluation in both experiments started at 85 days of gestation and ended at weaning of piglets (21 days old), with an average of 50 days of trial.

For the first test, 120 sows were used, subjected to an experimental design in randomized blocks in three groups by parity order (parity 1 and 2, parity 3 and 4, parity 5–6) of 40 sows per block. Within each block, sows were randomly distributed among the 4 treatments, with 30 replicates (sows), with the sow and its respective litter being the experimental units. In Experiment II, 200 sows were subjected to a 2 × 2 factorial design, divided into randomized blocks per parity order (parity 1 and 2, parity 3 and 4, parity 5–6), with the use of capsaicin and vitamin 1,25-(OH)_2_D_3_ as factors. There were 50 replicates per treatment in this experiment, with the sow and its litter being the experimental units.

In Experiment I, 120 sows were subjected to a randomized block design with four treatments: T1—control (without Vit 1,25-(OH)_2_D_3_ and supplemental capsaicin); T2—3.5 µg Vit 1,25-(OH)_2_D_3_/gestation/day and 7.0 µg Vit 1,25-(OH)_2_D_3_/lactation/day; T3—7.0 µg capsaicin/gestation/day and 14.0 µg capsaicin/lactation/day; T4—1.75 µg Vit 1,25-(OH)_2_D_3_ + 3.5 µg capsaicin/gestation/day and 3.5 µg Vit 1,25-(OH)_2_D_3_ + 7.0 µg capsaicin/lactation/day.

In Experiment II, 200 sows were randomly blocked, factorial 2 × 2 (without or with inclusion of Vit 1,25-(OH)_2_D_3_; and without or with inclusion of capsaicin), as follows: T1—control (without Vit 1,25-(OH)_2_D_3_ and supplemental capsaicin); T2—3.5 µg Vit 1,25-(OH)_2_D_3_/gestation/day and 7.0 µg Vit 1,25-(OH)_2_D_3_/lactation/day; T3—7.0 µg capsaicin/gestation/day and 14.0 µg capsaicin/lactation/day; T4—3.5 µg Vit 1,25-(OH)_2_D_3_ + 7 µg capsaicin/gestation/day; and 7.0 µg Vit 1,25-(OH)_2_D_3_ + 14.0 µg capsaicin/lactation/day.

In both experiments, during the gestation phase, the sows were housed individually in crates and fed once a day. In the lactation phase, the sows were housed at 110 days of gestation in individual farrowing crates and kept there until weaning. From farrowing to weaning, the dose of additives offered during the gestation phase was doubled, being administered at the first meal of the day at 7:30 a.m. In total, four food treatments were practiced daily, characterizing ad libitum food management in the lactation phase. The gestation and lactation feeds used in Experiments I and II are shown in Table 1.

From the beginning to the end of the farrowing, the following variables regarding the reproductive performance of the sows were analyzed: (1) duration and type of delivery (normal or dystocic), (2) use of oxytocin, (3) total born, (4) percentage of born alive, (5) percentage of stillbirths, (6) percentage of mummified, (7) percentage of piglets born weighing less than 0.900 kg, (8) total litter weight at birth, (9) individual weight of the piglet at birth, (10) daily feed intake of the lactating sow, and (11) amount of colostrum produced. Farrowing progress was considered normal or dystocic. Normal was defined as a farrowing process taking place without obstetrical intervention. Dystocic was defined as when no piglet was delivered for more than 60 min and a manual obstetric intervention was carried out. In this case, fetuses accessible by hand were carefully pulled out. When no piglets were in the birth canal during manual obstetric interventions and there was no uterine contraction, the use of oxytocin during farrowing was allowed. In order to predict the colostrum produced through the weight of the piglets at birth and 24 h later, the equation described by Devillers et al. [16] was used:CI = −217.4 + 0.217 × t + 1861019 × BW/t + BWb × (54.80 − 1861019/t) × (0.9985 − 3.7 × 10^−4^ × t_FS_ + 6.1 × 10^−7^ × t_FS_^2^),
where CI = colostrum intake from t_0_ (g); BW = actual body weight (kg); BWb = body weight at birth (kg); t = time elapsed from t_0_ (min); t_FS_ = interval between birth and first sucking (min).

For the analyses of the reproductive performance, the piglets were identified individually with ear tags and weighed at the following times: (1) birth, (2) 24 h after birth, (3) after litter standardization, 2nd day of age, (4) at 10 days of age, and (5) at weaning. The results of this weighing were expressed as the mean piglet weight, litter weight, and daily weight gain (DWG). The percentage of mortality 24 h after birth and the amount of colostrum ingested were also observed.

Immediately after the movement of the piglets for the uniformity of the litters, the presence or absence of diarrhea was observed and classified, according to the protocol of Liu [17], into 1—liquid feces, 2—creamy feces, 3—pasty feces, and 4—normal stools. For data analysis, the stall was considered the experimental unit, and if at least one animal had a score below 3, the stall was considered positive for diarrhea. The same evaluation was performed at the time of weaning.

In both experiments, 40 sows (10 per treatment), six hours after the beginning of parturition, were submitted to blood collection by puncture (5.0 mL syringe, 40 × 10 mm needle) of the vessels in the neck region to determine the levels of serum calcium, phosphorus, C-reactive protein, and cortisol. The samples were collected in vacutainer tubes without anticoagulant for biochemical analysis and subsequently centrifuged (1000 rpm, 15 min). Calcium and phosphorus levels were evaluated using colorimetric biochemical methodology in a semiautomatic reader (Mindray BA-88A, Londrina, Paraná, Brazil). C-reactive protein was determined using the agglutination methodology (Gold Analise, Londrina, Paraná, Brazil), and for cortisol the chemiluminescence method (Immulite 2000 Immunoassay System) was used.

The trial design of Experiment I involved casual blocks, and data were analyzed with ANOVA (GLM) with the block and treatment with Tukey’s test for mean comparison. The trial design of experiment II was a full factorial design with blocks (2 × 2), with two additives as factors in experiment II. The assumption of data normality was tested using the Ryan–Joiner test (a test similar to Shapiro–Wilk), and the null hypothesis expressed that the residual data presented a normal distribution and that data not normally distributed were transformed (X^2^ + 1 or log_x_) before analysis. For diarrhea and other parameters related to frequency (percentage of dystocia rate, oxytocin use, and piglets below 900 g), a Chi^2^ test was applied. Statistical analysis was performed using Minitab (Version 21.2.0). The differences were considered significant with a probability level of 95% (*p* < 0.05) and as a trend with a probability level of 90% (*p* ≤ 0.10). The parity order was considered a block for the statistical design. ANCOVA (analysis of covariance) was used to determine the reproductive performance of sows (Experiment I), adjusting/controlling for piglet number and weight (kg) variables (covariates).

## 3. Results

### 3.1. Reproductive Performance of Sows

For Experiment I, there was a difference in the dystocia rate (*p* = 0.017), where vitamin 1,25-(OH)_2_D_3_ + capsaicin treatment (both at half dose) resulted in a lower occurrence of complications. Compared to the control group, there was an increase of approximately 233.33% in the incidence of dystocia births (Table 2).

Sows fed diets based on vitamin 1,25-(OH)_2_D_3_ + capsaicin had a lower percentage of piglets with body weights (*p* = 0.044) below 900 g at birth, which reflected a higher individual body weight at birth (1.360 kg), differing from the other treatments (Table 2). The sows that received the treatments supplemented with vitamin 1,25-(OH)_2_D_3_ and vitamin 1,25-(OH)_2_D_3_ + capsaicin had higher colostrum production (*p* = 0.003), reflecting the higher consumption of this supplement by the piglets (*p* = 0.001), with a positive effect on weight gain 24 h after birth (*p* = 0.001) (Table 2).

### 3.2. Reproductive Performance of Sows after Litter Standardization

In Experiment I (Table 3), the average number of piglets per sow after uniformity of litters favored the control group (*p* = 0.024). However, piglets from sows that received vitamin 1,25-(OH)_2_D_3_ showed higher body weight 10 days after the litter standardization compared to the other treatments (*p* = 0.001), with a daily weight gain higher than the treatment that associates vitamin 1,25-(OH)_2_D_3_ and capsaicin.

At weaning, the highest weight was maintained for those from sows that received vitamin 1,25-(OH)_2_D_3_, which was higher than the control group (*p* = 0.033).

During lactation, sows with capsaicin supplementation had a higher feed intake (7.938 kg) than sows that consumed diets supplemented with vitamin 1,25-(OH)_2_D_3_ (Table 3), but there was no difference in the feed efficiency between the treatments.

### 3.3. Incidence of Diarrhea

In Experiment I, piglets born to sows that consumed diets supplemented with the evaluated additives had a lower (*p* = 0.001) incidence of diarrhea in the first week of life (Table 4).

### 3.4. Blood Analyses

The results of blood analysis of the sows in Experiment I did not differ between treatments (Table 5).

### 3.5. Reproductive Performance of Sows

Considering the total number of piglets born, in Experiment II (Table 6), vitamin 1,25-(OH)_2_D_3_ supplementation resulted in faster births (with 1 h less duration) compared to the control group (*p* = 0.001). For dystocia, the group supplemented with vitamin 1,25-(OH)_2_D_3_, with or without capsaicin, showed better results. Thus, when associated with capsaicin, a ratio of 8.16 versus 20.65% dystocia was observed, whereas, for the condition not associated with capsaicin, there was a ratio of dystocia of 8.60 versus 19.58%. With the shorter delivery time, the percentage of stillborn was also lower for the group of sows that received vitamin 1,25-(OH)_2_D_3_ (*p* = 0.026) (Table 6).

### 3.6. Reproductive Performance of Sows after Litter Standardization

For Experiment II (Table 7), the individual weights of piglets from sows that received capsaicin were better at the time of standardization (*p* = 0.001), a result that persisted until the 10th day of life, indicating better DWG in the period. However, there was interaction between the vitamin and capsaicin factors for piglet weight at 10 days of age and for piglet DWG in this interval (2nd to 10th day of life).

Piglets from sows that did not receive capsaicin but were supplemented with vitamin D_3_ had a better DWG (*p* = 0.018) and weight at this age (*p* < 0.001) in relation to piglets born from sows not supplemented with vitamin 1,25-(OH)_2_D_3_, respectively, 0.220 versus 0.232 kg, and 3.357 versus 3.444. Similarly, piglets born to sows that did not receive vitamin 1,25-(OH)_2_D_3_ and that were supplemented with capsaicin had better DWG (*p* < 0.001) and better weight at this age (*p* < 0.014) in relation to piglets born from sows not supplemented with capsaicin, respectively, 0.220 versus 0.232 kg, and 3.357 versus 3.503 kg. Considering the entire lactation period, the feed consumption was similar between the factors, but the feed efficiency was better for sows that received capsaicin. At weaning, there was no interaction between factors or significant differences between capsaicin and vitamin 1,25-(OH)_2_D_3_ supplementation for all parameters evaluated.

### 3.7. Incidence of Diarrhea

In Experiment II, the frequency of diarrhea was significantly lower in the first week of life in piglets born of sows that received capsaicin (Table 8), and the supplementation with capsaicin and vitamin 1,25-(OH)_2_D_3_ resulted in lower frequencies of weaning diarrhea. There was an interaction of vitamin and capsaicin factors in this digestive disorder in the first week (*p* = 0.025) and at weaning (*p* < 0.001). Regarding the weaning week, piglets from sows that received vitamin 1,25-(OH)_2_D_3_, associated (3.18 versus 9.52%) or not associated (3.67 versus 8.76%) to capsaicin, showed minor clinical signs of diarrhea.

### 3.8. Blood Analyses

In Experiment II, compared to sows in the control group, sows that received 1,25-(OH)_2_D_3_ showed a trend of higher serum calcium levels (*p* = 0.07) (Table 9). The other blood parameters did not differ among the treatments (*p* > 0.05).

## 4. Discussion

The use of vitamin 1,25-(OH)_2_D_3_ and capsaicin in combination, either with a half dose or higher doses, was effective regarding the reproductive performance in both experiments. The positive results of vitamin 1,25-(OH)_2_D_3_, administered alone or in combination with capsaicin, regardless of the dose, on the frequency of dystocic births are promising, considering that modern sows are, due to hyper-prolificity [18], more likely to experience birth dystocia [5].

These results are attributed to the mechanism of the direct action of vitamin 1,25-(OH)_2_D_3_ on the homeostasis of calcium and phosphorus, minerals that are associated with uterine contraction [19]. Blim et al. [5] found an increased calcium level at the onset of the fetal expulsion stage. Immediately after the end of parturition, there was a sustained increase in Ca, which was clearly above the levels prior to farrowing. These findings show that calcium is a crucial electrolyte for the parturition process and an important messenger in the myometrial contraction mechanism [5]. In addition, vitamin D_3_ plays an important role in binding to the vitamin D receptor and activating several interaction pathways, which results in an increase in calcium in the circulation [20]. These effects are confirmed by muscle strength during parturition, which is influenced by the vitamin D receptor, which is capable of modulating hormones involved in uterine contraction [5,21].

The anti-inflammatory and analgesic properties of capsaicin may also have contributed to the reduction in dystocia births. Pepper extract is one of the main TRPV1 receptor agonists, causing desensitization via nociceptor dysfunction, resulting in decreased pain perception [22], a common framework during the calving process.

Although the serum calcium level in this study did not differ in both experiments, the tendency towards a higher level of the circulating mineral may be related to better absorption and metabolism than vitamin D_3_ (cholecalciferol) provides [23]. In another study, an increase in calcium levels was reported when using vitamin 1,25-(OH)_2_D_3_ in monogastric diets [13].

The decrease in the parturition time of sows supplemented with vitamin 1,25-(OH)_2_D_3_ supports the hypothesis of an increase in circulating Ca^2+^ at the time of parturition, which provides a more potent contraction of the myometrium and a lower number of stillborn [24]. This discussion on the increase in serum calcium level is also corroborated by the results found in Experiment II (Table 6), in which sows treated with vitamin 1,25-(OH)_2_D_3_ and vitamin 1,25-(OH)_2_D_3_ + capsaicin had a lower incidence of complicated parturition, shorter duration of parturition, and lower incidence of stillbirths. These results are consistent with the findings of Elrod et al. [25]; in that study, sows that received a supplement based on calcium chloride in the diet had a shorter delivery time and a lower rate of stillbirths than those in sows who did not receive this supplement. In this scenario, Blim et al. [5] observed that a large proportion of sows have previously unknown deficiencies in electrolyte homeostasis (Ca, Mg, and P) around and during parturition, which is reflected by the large number of dystocic farrowings and the deficiencies in electrolyte concentrations during dystocia. Borges et al. [18] suggested that the longer the calving time, the greater the number of stillborn, and the main reason for this is the interruption of blood flow in the placenta, resulting in hypoxia in the later animals, which, even if they do not die, may become lethargic and nonviable.

In Experiment I, the lower rate of piglets born weighing less than 900 g from sows that received vitamin 1,25-(OH)_2_D_3_ + capsaicin in the diet (Table 2) may indicate a better efficiency of these diets in the final period of gestation. According to Yuan [25], birth weight depends on the amount of nutrients available in the body, which are transferred to piglets via the placenta. Thus, the benefits of this active ingredient, which is related to improving enzyme secretion and intestinal integrity through inhibiting gastrin release and stimulation of somatostatin [9], may support these results.

Such effects may also support the best results for the feed intake of sows (Experiment I) when they received capsaicin, reaching an average intake of 7.938 kg feed/day. This result is also important for lactating sows due to their high catabolism, which can lead to greater mobilization of their body reserves, with losses in the subsequent cycle [26]. This finding can be attributed to the effect of tolerance to endogenous heat sensation by the sow, which this alkaloid provides from the desensitization of TRPV receptors responsible for the sensation of pain and heat in the body [27]. A higher feed intake during lactation or attenuation of catabolism during this phase improves reproductive performance in the subsequent cycle [28].

The mechanism of action of vitamin 1,25-(OH)_2_D_3_ on the digestive response is attributed to the activation of sensors of the gastrointestinal tract, such as transient member 1 of subfamily V of the cation channel, in the potential of the transient receptor [22]. In Experiment I, the sows that received only vitamin D_3_ supplementation had a lower feed intake. However, because vitamin 1,25-(OH)_2_D_3_ promotes better intestinal absorption of Ca^2+^, which contributes to the promotion of the plasma level of the mineral [29], the improvement in the zootechnical performance of the piglets observed in this treatment is probably due to the effects of the increased vitamin level in the milk fat and protein contents [30].

The results indicate that the performance of the piglets (Experiments I and II) was effectively influenced by the dietary supplementation of the sows with vitamin 1,25-(OH)_2_D_3_ and the association of this supplementation with capsaicin. These gains may be attributed to higher colostrum intake and lower occurrence of diarrhea compared to animals in the control group. The higher intake of colostrum (evaluated only in Experiment I) is justified by the higher production of colostrum by sows fed vitamin 1,25-(OH)_2_D_3_) (Table 2). Vitamin receptors are present in more than 90% of body tissues and play an important role in regulating hormones, such as oxytocin, which plays an intrinsic role during delivery and in the ejection of colostrum [12].

In Experiment I, the best results found in the daily weight gain of the piglets in the first 24 h after birth (Table 2), in the weight at the time of uniformity, and at 10 days of life, persisting until weaning (Table 4), are identified with the observations of Devillers et al. [6], which demonstrated that satisfactory colostrum intake can have positive effects on piglet growth up to three weeks after weaning.

The reason why piglets from sows that consumed the control diet or were supplemented with capsaicin alone had a lower colostrum intake than piglets from sows treated with vitamin 1,25-(OH)_2_D_3_ and vitamin 1,25-(OH)_2_D_3_ + capsaicin (observed in Experiment I) is attributed to the greater duration of delivery and higher percentage of stillbirths within the control group, which led to losses in colostrum synthesis, an approach advocated by Langendijk and Plush [31].

Specifically, related to capsaicin and Vit 1,25-(OH)_2_D_3_, the best weights found after animal standardization and at 10 days of life observed in Experiment II (Table 7) correspond to the findings of Matysiak et al. [32], who worked with different plant extracts (carvacrol, cinnamaldehyde, and capsaicin) in the diets of lactating sows and observed an increased lactose concentration in milk, with positive effects on the performance and survival rate of piglets. It is possible that the greater body weight gain of the piglets that we observed is attributed to the higher milk production determined by this capsaicinoid, possibly due to the higher feed intake and use of dietary nutrients. Related to this effect, capsaicin positively influenced feed efficiency, showing a synergistic relationship with litter weight gain and sow feed consumption.

Capsaicin also has recognized antioxidant and anti-inflammatory activity, intervening beneficially in the course of enteric diseases, thus contributing to intestinal health [33] and the production of inflammatory molecules in peritoneal macrophages stimulated by lipopolysaccharides (LPS), showing significant inhibitory activity in the production of prostaglandin E_2_ (PGE-2) induced by LPS by peritoneal macrophages [8].

The beneficial effects of additives in pig diets have been widely verified in terms of reducing the occurrence of diarrhea and promoting the health of the gastrointestinal tract [34], as observed in both experiments. Along these lines, Liu et al. [35] evaluated different plant extracts (oleoresin capsicum, botanical garlic, and turmeric oleoresin) in pig diets and found effective activation of immune responses and expression of genes involved in tight junctions, indicating that both effects may increase intestinal mucosal immunity and function of the intestinal barrier. This finding is in line with the results of the present study, where we observed a reduction in the occurrence of diarrhea in piglets from sows that received capsaicin in their diet.

Capsaicin exerts recognized pharmacological effects with modulatory properties in the activation or inhibition of several biochemical and immunological pathways [36]; however, the piglets kept with sows that received vitamin 1,25-(OH)_2_D_3_ showed a similar response (Experiments I and II). This finding is attributed to the effect of vitamin 1,25-(OH)_2_D_3_ in relieving intestinal injury and inhibiting the intestinal immune response induced by pathogens, with an effect on the expression of tight junction proteins to maintain the integrity of the intestinal barrier [37].

Concomitantly, vitamin 1,25-(OH)_2_D_3_ has a potential physiological role in the regulation of immune responses, along with the vitamin D receptor, which is induced after the activation of pathogen recognition receptors and can induce the synthesis of antibacterial proteins [38], which promotes improvements in the general health of animals through various nongenomic mechanisms, such as protein expression, inflammation, oxidative stress, and cellular metabolism [19].

Dietary supplementation with plant extracts, vitamin 1,25-(OH)_2_D_3_, and capsaicin during the final third of gestation and lactation reduced the occurrence of dystocic births and decreased the incidence of diarrhea in litters. The isolated use of natural vitamin 1,25-(OH)_2_D_3_ increased colostrum production, decreased the calving time and number of stillbirths, and maintained the best piglet weight until weaning. The unique use of capsaicin led to higher feed intake and was also positive in the litter evolution. Thus, the use of additives improved the reproductive performance of sows and contributed to litter growth.

## 5. Conclusions

Vitamin 1,25-(OH)_2_D_3_ and capsaicin can be used as additives to improve the farrowing efficiency and colostrum production, with positive consequences for the performance of piglets until weaning, representing resources to improve the performance of high-prolific sows. Additionally, these additives are capable of minimizing diarrhea in suckling piglets. The evaluated doses and the association of additives must be considered in their use while considering the magnitude of their actions and the cost–benefit results.

## Figures and Tables

**Table 1 animals-13-02794-t001:** Dietary ingredients (%) and calculated nutritional values of the gestation and lactation diets used in Experiments I and II.

Ingredients	Experiment I		Experiment II	
	Gestation	Lactation	Gestation	Lactation
Corn grain (7.85%)	62.98	53.49	60.85	53.87
Soybean meal (45%)	13.52	27.68	12.04	25.48
Soybean hull	19.01	-	22.07	-
Meat meal (77.1%)	3.67	5.00	3.50	5.00
Wafer bran	-	5.00	-	5.00
Energy concentrate ^1^	-	7.50	0.40	-
Limestone (35.8%)	0.32	-	0.14	0.90
Sugar	-	-		5.00
Dicalcium phosphate (17% Ca/21%P)	-	-	-	1.50
L-Lisina 78.4%	-	0.40	-	0.40
DL-Methionine 99%	-	0.19	-	0.19
L-Tryptophan 99%	-	0.05	-	0.06
L-Threonine 98%	-	0.19	-	-
Salt	0.40	0.40	-	0.50
Soybean oil	-	-	-	2.00
Premix ^2^	0.10	0.10	1.00	0.10
Nutrients per kg of feed				
Metabolizable energy, kcal	3008	3683	3007	3353
Crude protein, %	14.69	21.01	14.42	19.28
Extract ethereal, %	3.66	5.7	3.24	4.88
Crude fiber, %	9.18	3.25	10.43	2.69
Calcium, %	1.00	1.01	0.87	0.95
Total phosphorus, %	0.45	0.62	0.45	0.56
Available phosphorus, %	0.40	0.51	0.40	0.36
Digestible lysine, %	0.57	1.20	0.64	1.06
Digestible threonine, %	0.74	0.67	0.46	0.68
Digestible methionine, %	0.33	0.37	0.23	0.26
Digestible tryptophan, %	0.16	0.23	0.14	0.20
Met + digestible cys, %	0.50	0.61	0.43	0.56

^1^ Energy Farms^®^ (De Heus Animal Nutrition). ^2^ Premix, guaranteed levels per kg of product: Choline: 75,000 mg/kg^−1^, Vitamin A: 50,000 IU, Vitamin D_3_: 75,000 IU, Vitamin E: 9000 mg/kg^−1^, Vitamin K_3_: 975 mg kg^−1^, Vitamin B_1_: 500 mg/kg^−1^, Vitamin B_2_: 1200 mg/kg^−1^, Vitamin B_6_: 750 mg/kg^−1^, Vitamin B_12_: 8000 mg/kg^−1^, Vitamin D: 3900 IU/kg; Niacin: 5000 mg/kg^−1^, Pantothenic acid: 3000 mg/kg^−1^, Folic acid: 500 mg/kg^−1^, Biotin: 20,000 mg/kg^−1^, Iron: 30,000 mg/kg^−1^, Copper: 3000 mg/kg^−1^, Manganese: 17,500 mg/kg^−1^, Zinc: 30,000 mg/kg^−1^, Iodine: 200 mg/kg^−1^, Selenium: 150 mg/kg^−1^, Phytase: 25,000 U kg^−1^.

**Table 2 animals-13-02794-t002:** Reproductive performance of sows supplemented with vitamin D_3_, capsaicin, or both (in half dosage) at 85 days of gestation until the end of lactation (Experiment I).

Parameters	Treatments				CV (%) *	*p*-Value
	Control	Vitamin 1,25-(OH)_2_D_3_	Capsaicin	Vit. 1,25-(OH)_2_D_3_ + Capsaicin		
Time of delivery, hours	5:45	5:00	5:40	5:05	63.61	0.843
Dystocia rate, %	40 ^a^	23 ^ab^	23 ^ab^	12 ^b^	-	0.017 ^2^
Oxytocin use, %	23	10	10	07	-	0.310
Total births, %	17.3	17.3	16.7	16.2	21.12	0.563
Live births, *n*	15.4	15.5	15.5	14.5	21.63	0.679
Stillbirths, %	7.5	5.9	4.6	3.3	123.84	0.120
Mummification, %	3.6	4.8	2.6	6.4	144.47	0.261
Piglets below 900 g, %	14.9 ^a^	13.9 ^ab^	14.4 ^ab^	10.3 ^b^	-	0.044 ^2^
Litter weight, kg	19.0	20.8	19.7	19.8	22.97	0.557
Average weight at birth, kg	1.28 ^b^	1.33 ^a^	1.27 ^b^	1.36 ^a^	16.22	0.001 ^1^
CV at birth, % *	20.77	21.22	21.87	18.32	31.21	0.196
Piglet weight at 24 h, kg	1.37 ^b^	1.45 ^a^	1.36 ^b^	1.47 ^a^	16.55	0.001 ^1^
Weight litter at 24 h, kg	19.3	21.6	20.8	20.8	76.05	0.344
Piglet daily weight gain 24 h, g	95 ^b^	139 ^a^	95 ^b^	121 ^a^	96.97	0.001 ^1^
Colostrum intake, g *	258 ^b^	330 ^a^	254 ^b^	312 ^a^	36.54	0.001 ^1^
Colostrum production, kg *	3772 ^b^	4906 ^a^	3810 ^b^	4402 ^a^	33.69	0.003 ^1^
Mortality to 24 h, %	5.1	4.7	5.2	4.2	12.42	0.534

¹ ^a, b^ Distinct letters in the rows indicate differences (*p* < 0.05) and trends (*p* < 0.10) determined using Tukey’s test. ^2 a, b^ Means followed by different letters in the rows indicate differences determined using the chi-square test (*p* < 0.05). * CV: coefficient of variation.

**Table 3 animals-13-02794-t003:** Reproductive performance of sows supplemented with vitamin 1,25-(OH)_2_D_3_, capsaicin, or both (in half dosage) from 85 days of gestation until the end of lactation, according to the evaluation time (days) (Experiment I).

Age	Parameters	Treatments				CV (%) *	*p*-Value
		Control	Vitamin 1,25-(OH)_2_D_3_	Capsaicin	Vit. 1,25-(OH)_2_D_3_ + Capsaicin		
2 days	Number of piglets, *n*	14.77 ^a^	14.03 ^b^	14.63 ^a^	14.58 ^a^	6.89	0.024
	Average birth weight, kg	1.55 ^b^	1.68 ^a^	1.49 ^b^	1.54 ^b^	28.52	0.001
	Average litter weight, kg	22.72	23.74	21.90	22.63	20.42	0.509
10 days	Average piglet weight, kg	2.974 ^b^	3.192 ^a^	3.043 ^b^	2.997 ^b^	24.08	0.001
	Average litter weight, kg	41.5	43.10	42.48	40.87	19.80	0.758
	Daily weight gain, g	216 ^ab^	227 ^a^	218 ^ab^	206 ^b^	33.66	0.001
Weaning	Number of piglets, *n*	13.87	13.48	13.83	13.54	0.070	0.102
	Average birth weight, kg	6.10 ^b^	6.36 ^a^	6.20 ^ab^	6.167 ^ab^	21.20	0.033
	Average litter weight, kg	81.14	83.22	81.79	78.67	16.48	0.656
	Daily weight gain, g	253	261	262	260	24.71	0.161
	Mortality rate, %	5.94	4.80	5.65	7.16	38.08	0.535
Daily feed intake during the lactation, kg/d		7.68 ^ab^	6.92 ^b^	7.94 ^a^	7.01 ^ab^	13.35	0.038
Feed efficiency, kg piglet/kg feed		0.439	0.490	0.443	0.438	27.86	0.679

^a,b^ Means followed by different letters in the rows indicate differences according to Tukey’s test (*p* < 0.05). * CV: coefficient of variation.

**Table 4 animals-13-02794-t004:** Percentage of diarrhea in piglets of sows supplemented with vitamin D_3_, capsaicin, or both (in half dosage) at 85 days of gestation until the end of lactation (Experiment I).

Ages	Treatments				*p*-Value
	Control	Vitamin 1,25-(OH)_2_D_3_	Capsaicin	Vit. 1,25-(OH)_2_D_3_ + Capsaicin	
First week, %	15.09 ^b^	6.99 ^a^	6.95 ^a^	6.71 ^a^	0.001
Weaning, %	3.21	1.40	3.21	2.52	0.503

^a,b^ Means followed by different letters in the rows indicate differences determined using the chi-square test (*p* < 0.05).

**Table 5 animals-13-02794-t005:** Blood analysis of sows supplemented with 1,25-dihydroxyvitamin D_3_, capsaicin, or both (half dosage) at 85 days gestation until the end of lactation (Experiment I).

Parameter	Treatments				CV (%) *	*p*-Value
	Control	Vitamin 1,25-(OH)_2_D_3_	Capsaicin	Vit. 1,25-(OH)_2_D_3_ + Capsaicin		
Calcium, mg/dL	90.2	98.8	92.1	94.9	14.20	0.276
Phosphorus, mg/dL	69.4	76.5	75.6	76.1	20.39	0.344
C-reactive protein, mg/dL	8.8	11.1	9.6	9.4	51.50	0.792
Cortisol, mg/dL	75.7	45.9	38.3	46.7	69.91	0.183

* CV: coefficient of variation.

**Table 6 animals-13-02794-t006:** Reproductive performance of sows supplemented with vitamin D_3_, capsaicin, or both at 85 days of gestation until the end of lactation (Experiment II).

Parameters	Without Vit 1,25-(OH)_2_D_3_ and Capsaicin	Vit 1,25-(OH)_2_D_3_ without Capsaicin	Capsaicin without Vit 1,25-(OH)_2_D_3_	Vit 1,25-(OH)_2_D_3_ + Capsaicin	CV (%) *	*p*-Value		
						Vitamin	Capsaicin	V × C **
Delivery, hours	4:57	3:48	4:24	4:26	52.81	0.001	0.857	0.550 ^1^
Dystocia, %	27.63	4.21	13.46	11.95	-	0.001	0.752	0.002 ^2^
Oxytocin use, %	6.59	3.12	6.81	3.37	-	0.287	0.316	0.592 ^2^
Total births, *n*	16.55	16.00	16.03	16.54	26.16	0.273	0.542	0.882 ^1^
Live births, *n*	14.60	14.51	14.16	15.07	26.81	0.615	0.138	0.519 ^1^
Stillbirths, %	7.61	5.37	6.92	5.92	124.11	0.026	0.306	0.168 ^1^
Mummification, %	3.58	3.27	4.06	2.74	184.21	0.706	0.162	0.565 ^1^
Litter weight, kg	19.52	19.70	19.02	20.27	24.74	0.878	0.080	0.290 ^1^
Average weight at birth, kg	1.36	1.36	1.37	1.35	15.12	0.842	0.587	0.208 ^1^
CV at birth, %	20.09	19.70	19.76	20.05	29.64	0.575	0.716	0.578 ^1^

^1^ Means followed by different letters in the rows indicate differences according to Tukey’s test (*p* < 0.05). ^2^ Means followed by different letters in the rows indicate differences found using the chi-square test (*p* < 0.05). * CV: coefficient of variation. ** V × C represents the vitamin 1,25-(OH)_2_D_3_ × capsaicin interaction.

**Table 7 animals-13-02794-t007:** Reproductive performance of sows supplemented with vitamin 1,25-(OH)_2_D_3_, capsaicin, or both at 85 days of gestation until the end of lactation, according to the evaluation time (days) (Experiment II).

Age	Parameters	Without Vit 1,25-(OH)_2_D_3_ and Capsaicin	Vit 1,25-(OH)_2_D_3_ without Capsaicin	Capsaicin without Vit 1,25-(OH)_2_D_3_	Vit 1,25-(OH)_2_D_3_ + Capsaicin	*p*-Value			
						CV (%) *	Vitamin	Capsaicin	V × C **
2 days	Number of piglets, *n*	13.64	13.54	13.43	13.78	12.61	0.785	0.307	0.471
	Average birth weight, kg	1.62	1.61	1.59 ^b^	1.64 ^a^	19.73	0.606	0.001	0.519
	Average litter gain, kg	21.83	21.91	21.39	22.41	23.92	0.940	0.275	0.758
10 days	Average piglet weight, kg	3.44	3.45	3.40 ^b^	3.49 ^a^	16.28	0.175	0.004	0.025
	Average litter weight, kg	44.06	44.60	43.13	45.71	21.29	0.807	0.115	0.407
	Daily weight gain, g	238	229	232b	235a	29.53	0.062	0.002	0.003
Weaning	Number of piglets, *n*	12.64	12.62	12.52	12.76	13.87	0.976	0.450	0.390
	Average piglet weight, kg	6.05	6.10	6.01	6.14	15.14	0.977	0.342	0.968
	Average litter weight, kg	76.19	76.68	74.88	78.17	17.94	0.861	0.235	0.655
	Daily weight gain, g	247	248	244	251	28.20	0.944	0.330	0.942
	Mortality rate, %	9.48	6.50	8.47	7.34	50.78	0.100	0.615	0.336
Daily feed intake, kg/d		6.41	6.42	6.45	6.37	14.82	0.294	0.625	0.400
Feed efficiency, kg piglet/kg feed		0.55	0.53	0.52	0.56	33.84	0.380	0.046	0.410

^a,b^ Means followed by different letters in the rows indicate differences according to Tukey’s test (*p* < 0.05). * CV: coefficient of variation. ** V × C represents the interaction vitamin 1,25-(OH)_2_D_3_ × capsaicin.

**Table 8 animals-13-02794-t008:** Percentage of diarrhea in piglets of sows supplemented with vitamin D_3_, capsaicin, or both at 85 days of gestation until the end of lactation (Experiment II).

Ages	Without 1,25-(OH)_2_D_3_ and Capsaicin	Vit 1,25-(OH)_2_D_3_ without Capsaicin	Capsaicin without Vit 1,25-(OH)_2_D_3_	Vit 1,25-(OH)_2_D_3_ + Capsaicin	*p*-Value		
					Vitamin	Capsaicin	V × C *
First week, %	11.63	10.03	12.10	9.50	0.284	0.039	0.025
Weaning, %	3.60	1.55	3.61	1.54	0.001	0.001	0.001

* V × C represents the interaction vitamin 1,25-(OH)_2_D_3_ × capsaicin.

**Table 9 animals-13-02794-t009:** Blood analysis of sows supplemented with 1,25-dihydroxyvitamin D_3_, capsaicin, or both (half dosage) at 85 days gestation until the end of lactation (Experiment II).

Parameter	Without Vit D_3_ and Capsaicin	Vit D_3_ without Capsaicin	Capsaicin without Vit D_3_	Vit D_3_ + Capsaicin	*p*-Value			
					CV (%) *	Vitamin	Capsaicin	V × C **
Calcium, mg/dL	93.1	88.5	91.6	90.7	10.39	0.07	0.62	0.96
Phosphorus, mg/dL	68.9	67.4	67.2	68.8	11.20	0.62	0.71	0.29
C-reactive protein, mg/dL	6.4	6.6	6.3	6.5	34.07	0.82	0.57	0.56
Cortisol, mg/dL	53.2	43.2	52.8	43.6	64.60	0.56	0.57	0.24

* CV: coefficient of variation. ** V × C represents the interaction vitamin 1,25-(OH)_2_D_3_ × capsaicin.

## Data Availability

The datasets generated during and/or analyzed during the current study are available from the corresponding author upon reasonable request.

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
