# Peer review of "Capsaicin and 1,25-Dihydroxyvitamin D_3_ Glycoside: Effects on the Reproductive Performance of Hyper-Prolific Sows"

_animals, 2023, doi:10.3390/ani13172794_

Round 1
Reviewer 1 Report
General comments:
The authors have stated (L 69 to 73) that Solanum glaucophyllum plants contain a glycosylated 1,25-dihydroxyvitamin D3. However, they have chosen to abbreviate the metabolite as vitamin D3. This abbreviation, used throughout the text, is confusing and misleading. The abbreviation must be changed to reflect that the metabolite is the active compound. I suggest 1,25 (OH)2 D3 or a similar abbreviation that includes 1, 25….. -. If vitamin D3 is assumed, the diets described in the abstract would be perceived as deficient in vitamin D, however, if abbreviated as 1,25 (OH)2 D3, nutritionists would conclude that diets were sufficient in vitamin D. An example of the confusion is illustrated in L84, does vitamin D3 refer to tissue concentrations of 1,25 (OH)2 D3, 25-OH D3 (the most abundant metabolite stored in tissues) or D3?
From the description of the control treatment 1 (L109 -111, and Table 1 footnotes), the amount of vitamin D metabolites added is not clear. Was no vitamin D, regardless of the metabolite source, included in treatment 1, or was the D3 metabolite added?
Specific Comments:
Table 1. Correct spelling of lysine
L 130 to 131. Why are 2 sources of vitamin D listed? Are these different metabolites.
L 173. Define DOE
Table 5. Correct the units for Ca and P. These should be mg/dL. Check the units for C-reactive protein and Cortisol. Also Table 9.
L 326-327. The specific metabolite of vitamin D is not clear if vitamin D3 is defined as an abbreviation for 1,25 (OH)2 D3
minor edits are required.
Author Response
Thank you sincerely for dedicating your time and expertise to review our article titled "Capsaicin and 1,25-dihydroxyvitamin D3 glycoside: effects on the reproductive performance of hyperprolific sows." We genuinely appreciate the valuable insights and constructive comments you provided to enhance the quality of our work.
We would like to inform the reviewers that all the requested changes have been highlighted in yellow in the revised version of the manuscript. We have carefully incorporated the suggested improvements and believe that these enhancements have strengthened the overall quality and clarity of our work. We are grateful for the valuable feedback provided by the reviewers, as it has been instrumental in refining our article. Thank you for your continued support and guidance in improving our research.

Reviewer 2 Report
1. In this study, it was observed that capsaicin improved the colostrum yield of sows and thus promoted the growth performance of piglets. However, no colostrum was detected in the experiment
2. How is the amount and pattern of capsaicin and vitamin D3 supplementation determined?
3. It is recommended to indicate which dosage and mode of addition is optimal
good work
Author Response
Dear Reviewer,
Thank you sincerely for dedicating your time and expertise to review our article titled "Capsaicin and 1,25-dihydroxyvitamin D3 glycoside: effects on the reproductive performance of hyperprolific sows." We genuinely appreciate the valuable insights and constructive comments you provided to enhance the quality of our work.
We would like to inform the reviewers that all the requested changes have been highlighted in green in the revised version of the manuscript. We have carefully incorporated the suggested improvements and believe that these enhancements have strengthened the overall quality and clarity of our work. We are grateful for the valuable feedback provided by the reviewers, as it has been instrumental in refining our article. Thank you for your continued support and guidance in improving our research.

Reviewer 3 Report
Simple summary.
Line 12 too complex sentence. Simplify.
Line 15. Rewrite section “…or not”.
Line 17. By “delivery” do you mean farrowing duration or parturition?
Line 17. Dystocia rate decreased that received.. please rewrite
Line 18. Higher not greater. Sows which received vit d3 had a higher production of colostrum? Also, does this mean the control group did not receive d3?
Line 18. Input or findings? Rewrite sentence.
Line 21. Last stage of gestation?
Line 26. Were those the same sows as in experiment 1? What about parity structure?
Line 34. Resulting
Keywords pepper?
Line 41. What this word contemporary means in this context? Wouldn’t it be clearer if you wrote lower birth weight along with less uniform?
Line 42. Which females? This requires rewriting. Sentence is unclear. How is a glucose level connected to anything? It was written without context. No previous or the next sentence help the reader.
Line 46. How do you define hyperprolific sows? This word is used throughout the manuscript without proper definitions. Todays sows are all hyperprolific. That is what the industry demands from breeding companies – larger litter size. The only exceptions are sire lines and traditional breeds. Please be more specific.
Line 47. This is actually the most important sentence in this manuscript, please put your focus on this.
Line 49. Where is dystocia defined?
Line 50. It is not a consequence of hyperprolific sows. Rewrite.
Line 52. Predisposing whom? Sows or piglets?
Line 54. Again starting the sentence with recognizing..?
Line 54. Are modern or hyperprolific sows? Be consistent with wording.
Line 61. I recommend not start paragraphs and/or sentences with a verb.
Line 66. Double space after comma?
Line 77. Why so many short paragraphs? And why a detailed description of Solanum glaucophyllum but not the other plants? Either all or none. What is the important bioactivity here? There is no supporting sentence to explain to the reader.
Overall. The flow is missing. It was more read as the commercial product advertisement rather than a scientific paper. This requires more work.
Materials and methods.
Line 91. Why is here the word conversely?
Line 95. Given that you used danbred and Topigs sows, did you ask for the approval from both companies to publish these figures? Those were not danbreed/Topigs genotypes. Were both farms populated with sows from both companies or? I do not see any of the authors being from these companies nor I see anywhere written that both companies agree to publish their data. In addition, are you certain that the company is called Topigs and not Topigs Norsvin?
No indication of the health status of the sows prior or during the experiment. What was the treatment used? More information required.
Line 98. What is the maternity sector?
Line 99. Was started? That is Grammarly incorrect sentence.
Line 123. Why “The sows”?
Line 137. What is the total number of births? Total piglets born or?
Line 138. Mumification is a process.
Line 139. Why 0.9 kg? why not 0.8 or 0.7?
Line 139. Individual weight of 1 piglet in litter or all piglets?
Line 141. Farrowing process was not divided. It was defined based on some criteria which is described later.
Line 141. No manual intervention can also happen when no one checks on the sows. Not necessarily because help was not required. Was there farm staff at all time next to the sows during the farrowing process? Hard to imagine this without cameras.
Line 144. Piglets were not removed. Reword.
Line 146. Do I need to read the paper by Devillers on the methodology or will that be described elsewhere in the paper?
Line 149. What is the zoothechical performance?
Line 150. Where were piglets tagged?
Line 150. So each piglet was put on the scale immediately after being born? What about night farrowings? In addition, exactly 24 hours after again?
Line 151. What is after uniformity?
Line 155. Soon after is not an expression for a scientific paper.
Were piglets moved to the trial sows or? What was the criterion for moving piglets?
Line 161. 40 females as sows or 40 female piglets? Please avoid using the word females, it makes it very hard to understand what are you referring to.
Line 177. Diarrhoea and other analyses?
Line 178. What kind of correction?
Line 190. No space between text and the table.
Daily weight gain of piglet or litter?
Table doesn’t say units in brackets, it is not aligned and it doesn’t specify if its on the piglet or litter level
3.2. it was sows, then females and now swine. This is very very confusing
What is litter index?
Line 207. In favour rewrite
Line 209. Too long sentence, needs rewriting and it is a higher daily gain, not better.
Please use subscript in tables
Is this average feed intake throughout the entire lactation?
Line 222. Breeders consumed diet?
3.5. again swine?
Line 238. What is a shorter birth?
Is the farrowing duration relative to the total number of piglets born? That has a huge effect
Line 241. What is an interaction effect? Between what?
Line 242, associated or not? Please rewrite.
Line 247. It is not a better litter weight.
3.6. now you use word sows?
Line 261. Relative to.
Line 266. What do you mean by all lactation periods? And what was the lactation length?
Line 277. What is the isolated effect?
Line 279. How can you know if piglets were moved from their dams to some other sows?
Line 302. The main rule in a discussion is to discuss your main findings first. Is your main finding that you did not find anything else? And can you claim this with absolute certainty or this is based on your knowledge?
Line 327. If you mention other authors, then surely there is more than 1 reference?
In a discussion you used few references and yet your opening statement in a discussion is that there aren’t references?
Line 347. There are many factors influencing piglet birth weight. You mentioned one of.
Line 378. In this sense is not an opening of a paragraph.
Line 385. This is correct, but it is not the only reason. There are other factors influencing reduction in feed intake.
Line 434. Avoid using words such as outstanding.
Conclusion section overuse of words e.g. promising, outstanding etc. without actually providing a conclusion. More information required, what is the recommendation? What is the expectation? Will this increase cost of production? How would you apply the product – in the feed or manually feeding? If latter, what about labour shortage globally?
must be improved
Author Response
Dear Reviewer,
Thank you sincerely for dedicating your time and expertise to review our article titled "Capsaicin and 1,25-dihydroxyvitamin D3 glycoside: effects on the reproductive performance of hyperprolific sows." We genuinely appreciate the valuable insights and constructive comments you provided to enhance the quality of our work.
We would like to inform the reviewers that all the requested changes have been highlighted in blue in the revised version of the manuscript. We have carefully incorporated the suggested improvements and believe that these enhancements have strengthened the overall quality and clarity of our work. We are grateful for the valuable feedback provided by the reviewers, as it has been instrumental in refining our article. Thank you for your continued support and guidance in improving our research.
Also the manuscript, after our revision, according your comments, was submitted to MDPI for English editing

Round 2
Reviewer 3 Report
This study investigated the effect of a natural source of Vit D3 and capsaicin in the late gestation and lactation on sows’ and piglets’ health as well as sows’ reproductive performance.
The authors have put significant effort into this manuscript. Good work.
Simple summary was too detailed in description of dosages. Not necessary to change it, but if there would be an avenue to focus on results and conclusions, it would improve readability.
Introduction started describing the impact on prolonged farrowing duration and depletion of calcium. This is in pig production to some extent neglected topic thus requires to be discussed more. Not in this manuscript per se than in general.
The introduction is excellent. The lacking part is a clear hypothesis. At this point, the reader will guess the reproductive performance shall improve, however, this needs to be clearly stated.
Materials and methods section very clearly described.
Was oxytocin given to all sows, irrespective of the treatment?
I am wondering, would it be clearer if the experiments are put in a table? Then in the text, just to refer to the group?
Discussion gives a good overview, literature has been used to support the results. The part with explanations of some results e.g. lower colostrum intake very well written.
Conclusions to the point.
Author Response
This study investigated the effect of a natural source of Vit D3 and capsaicin in the late gestation and lactation on sows’ and piglets’ health as well as sows’ reproductive performance.
Rev: The authors have put significant effort into this manuscript. Good work.
Rep: Thank you so much for taking the time to review our manuscript and for your kind words. We appreciate your recognition of the significant effort we put into this work. Your suggestions have been incredibly helpful in improving the quality of our paper. We sincerely appreciate your contribution and support.
Rev: Simple summary was too detailed in description of dosages. Not necessary to change it, but if there would be an avenue to focus on results and conclusions, it would improve readability.
Rep: We have taken your suggestion to heart and revised the simple summary to include more results and conclusions.
Rev: Introduction started describing the impact on prolonged farrowing duration and depletion of calcium. This is in pig production to some extent neglected topic thus requires to be discussed more. Not in this manuscript per se than in general.
Rep: Thank you for your feedback on our introduction. We agree that the impact of prolonged farrowing duration and depletion of calcium is an important topic in pig production that requires further discussion. We appreciate your suggestion to explore this topic in more detail in future research.
Rev: The introduction is excellent. The lacking part is a clear hypothesis. At this point, the reader will guess the reproductive performance shall improve, however, this needs to be clearly stated.
Rep: Thank you for highlighting this point in our manuscript. We have revised the introduction to include a clear statement of the study's hypothesis, as suggested by the reviewer. The changes were made in lines 81 to 85 to improve the clarity of the hypothesis.
Rev: Materials and methods section very clearly described.
Rep: Thank you for your feedback on the materials and methods section. We appreciate your comments.
Rev: Was oxytocin given to all sows, irrespective of the treatment?
Rep: No, oxytocin was not given to all sows indiscriminately. The use of oxytocin during farrowing was allowed only when there were no piglets in the birth canal during manual obstetric interventions and no uterine contractions were present. Lines 153 to 155.
Rev: I am wondering, would it be clearer if the experiments are put in a table? Then in the text, just to refer to the group?
Rep: Thank you for your feedback. We appreciate your time and effort. We understand your suggestion to put the experiments in a table. We tried this approach, but we found that it made the text difficult to read and understand. The experimental design of our study is complex, and we believe that it is important to provide a detailed description of each experiment in the text. We have revised the article to make it clear that the experiments are presented separately. We have also added a table that summarizes the main findings of each experiment. We believe that this approach provides a clear and concise overview of the study.
Rev: Discussion gives a good overview, literature has been used to support the results. The part with explanations of some results e.g. lower colostrum intake very well written.
Rep: Thank you for your positive feedback on the discussion section. We appreciate your comments.
Rev: Conclusions to the point.
Rep: Thank you for your valuable feedback.
